# Multiple Concurrent Causal Relationships and Multiple Governance Pathways for Non-Pharmaceutical Intervention Policies in Pandemics: A Fuzzy Set Qualitative Comparative Analysis Based on 102 Countries and Regions

**DOI:** 10.3390/ijerph20020931

**Published:** 2023-01-04

**Authors:** Yaming Zhang, Jiaqi Zhang, Yaya Hamadou Koura, Changyuan Feng, Yanyuan Su, Wenjie Song, Linghao Kong

**Affiliations:** 1School of Economics and Management, Yanshan University, Qinhuangdao 066004, China; 2Internet plus and Industrial Development Research Center, Yanshan University, Qinhuangdao 066004, China; 3School of Foreign Languages, Yanshan University, Qinhuangdao 066004, China; 4Business School, University of Granada, Campus Universitario de Cartuja, 18071 Granada, Spain

**Keywords:** viral transmission, non-pharmaceutical intervention, grounded theory, COVID-19, fsQCA, SEIQR

## Abstract

The global outbreak of COVID-19 has been wreaking havoc on all aspects of human societies. In addition to pharmaceutical interventions, non-pharmaceutical intervention policies have been proven to be crucial in slowing down the spread of the virus and reducing the impact of the outbreak on economic development, daily life, and social stability. However, no studies have focused on which non-pharmaceutical intervention policies are more effective; this is the focus of our study. We used data samples from 102 countries and regions around the world and selected seven categories of related policies, including work and school suspensions, assembly restrictions, movement restrictions, home isolation, international population movement restrictions, income subsidies, and testing and screening as the condition variables. A susceptible-exposed-infected-quarantined-recovered (SEIQR) model considering non-pharmaceutical intervention policies and latency with infectiousness was constructed to calculate the epidemic transmission rate as the outcome variable, and a fuzzy set qualitative comparative analysis (fsQCA) method was applied to explore the multiple concurrent causal relationships and multiple governance paths of non-pharmaceutical intervention policies for epidemics from the configuration perspective. We found a total of four non-pharmaceutical intervention policy pathways. Among them, L1 was highly suppressive, L2 was moderately suppressive, and L3 was externally suppressive. The results also showed that individual non-pharmaceutical intervention policy could not effectively suppress the spread of the pandemic. Moreover, three specific non-pharmaceutical intervention policies, including work stoppage and school closure, testing and screening, and economic subsidies, had a universal effect in the policies grouping for effective control of the pandemic transmission.

## 1. Introduction

The COVID-19 virus spread has been unprecedently exponentially reaching over 224 countries and territories worldwide in a very short period. In the absence of medical solutions such as vaccines and effective drugs, governments have implemented non-pharmaceutical intervention policies to delay and contain this rapid propagation of COVID-19 virus in the population, delay the pandemic peak, and reduce casualties [1,2]. Non-pharmaceutical interventions were additionally used to slow the spread of the virus prior to the mass production of vaccines to reduce the epidemic’s impact on the economy, people’s daily lives, and social stability [3,4]. Numerous non-pharmaceutical policies have been implemented around the world. Finding the right “combination” of these policies and interventions to deal with the spread of pandemics is a very important social development problem [5,6]. In reality, the adopted non-pharmaceutical intervention policies and their effectiveness in mitigating the pandemic effects vary from country to country. It is still unclear what non-pharmaceutical intervention policies are more effective to curb the spread of a pandemic. This issue plagues both academia and practitioners. Opening the “black box” of non-pharmaceutical intervention policies and pandemic prevention and control mechanism will help decision makers and respective authorities to formulate and adjust pandemic prevention and control policies more scientifically and rationally.

Qualitative comparative analysis (QCA) insists on the configuration perspective, changing the traditional “qualitative” or “quantitative” single-analysis idea by integrating the advantages of the two methods to explore the multiple concurrent causal relationships between complex factors. By integrating the advantages of both methods, we explore the multiple concurrent causal relationships between complex factors and open up new research ideas and analysis methods based on conditional histories [7]. This article analyzes how non-pharmaceutical intervention policies prevent and control the spread of pandemics from the configuration perspective. Using the fsQCA method to analyze data from a sample of 102 countries and regions around the world and combining existing literature results and perspectives, seven representative conditional variables were selected to explore the conditional histories and pathways of non-pharmaceutical intervention policies with effective control of pandemic spread.

The rest of the paper is organized as follows: In Section 2, we review the current literature on non-pharmaceutical intervention policies for pandemics. In Section 3, we discuss the research methodology and design and identify the selection and calculation of research variables. In Section 4, we describe data sources and the selection of the corresponding case sample. In Section 5, we provide an empirical analysis of the selected data and discuss the non-pharmaceutical intervention policies grouping construction. In Section 6, we discuss this investigation’s findings and conclusions.

## 2. Literature Review

To date, what non-pharmaceutical intervention policy can effectively control the spread of the pandemic is an urgent problem. For this topic, scholars have mainly studied aspects of a single policy from the quantitative analysis.

Most studies have focused on social distance policies such as work stoppage, home quarantine, and movement restrictions. Capon et al. found that implementing a school closure policy continued to be effective until the outbreak was largely contained [8]. Walsh et al. found that a short suspension led to a temporary alleviation of the outbreak, but once the policy was lifted, the outbreak peaked again [9]. A reactive strategy of implementing school closures when a certain percentage of students become symptomatic can reduce the severity of an outbreak [10]. In addition to schools, studies have shown that workplace contacts account for 20–25% of all contacts and 16% of all transmitters [11], and workplace disease infection rates exceed 20% in severe pandemics [12], while social distance policies in workplace reduce the total number of flu cases and delay the flu peak [13]. Factors such as the length of time people gather or travel, the proportion of trips, and the initial reproduction number can also influence the spread of a pandemic, and postponing or canceling large public gatherings can be effective in curbing pandemics [14]. Limiting large gatherings and other measures to maintain social distance may help in reducing transmission [15,16]. The spread of the epidemic is influenced by consumption motives [17]. When social distance exceeds 1 m, it can reduce virus infection to a great extent, and the probability of contracting the virus will be halved for every 1 m increase in social distance within a certain range [18]. Reducing population density at fixed sites can reduce the rate of virus transmission [19,20]. Public transport plays an important role in disease transmission and control [21,22]. Some studies found that the implementation of combined isolation and tracing strategies significantly reduced the size and number of infections [23,24], and that earlier implementation of international travel restriction policies reduced COVID-19 mortality by 62% [25], and targeted implementation of travel restriction policies will still achieve better outbreak prevention and control results [26]. Steyn N’s team found through epidemiological modeling analysis that contact isolation rates of up to three-quarters of the total population could delay outbreaks [27].

Some previous studies have estimated that a home isolation policy in the early stages of an outbreak could reduce mortality by about 50% [28,29], while another study showed that the number of deaths would be 167% higher without a home isolation policy [30]. The results of 10 experimental simulations of the spread of COVID-19 virus, using Italy and the Diamond Princess cruise ship as experimental samples, all showed that centralized isolation measures not only led to a significant reduction in infection and mortality rates but also resulted in reduced pandemic prevention expenditures [31]. In addition, the mortality rate of COVID-19 did not change significantly after adjusting for other non-pharmaceutical intervention policies by controlling variables, indicating a low degree of interaction between home isolation policies and other policies. Most of these studies have analyzed the influence of a non-pharmaceutical intervention policy on the effectiveness of pandemic prevention and control, ignoring the intersection and interaction among variables of the pandemic determinant. Nevertheless, how non-pharmaceutical intervention policies affect pandemic prevention and control is a complex process consisting of multiple elements interacting and jointly influencing each other, and we estimate that the internal relationship between the elements needs to be explored from a systematic or holistic perspective. We consider that including multiple interactions of various factors affecting the effectiveness of epidemic prevention and control will render the results more realistic.

In addition to studies of social isolation policies, scholars have also conducted in-depth studies of economic subsidy policies as well as testing and screening policies. Allen et al. analyzed the effectiveness of a short-term home isolation access debt-relief policy implemented in Canada in response to the coronavirus pandemic [32]. Farzanegan found that workplace population mobility rates decreased by 4.4–8.29%, a 21.8–47.7% reduction in the rate of COVID-19 cases three weeks after the implementation of the income subsidy policy, and a 17.1–29.7% reduction in the rate of mortality after five weeks [33]. In addition, Miyawaki found that countries with income subsidy levels of 50% of the pay scale had a more pronounced suppression of the pandemic spread [34]. It can be seen that the mechanism of action of income subsidy policies is different from that of social distance type policies, as the former are complementary in that they increase the willingness of people to comply with the relevant policy measures through encouragement, while the latter are complementary in that they achieve the goal of controlling the pandemic spread through mandatory regulations. By studying the number of confirmed diagnoses and deaths in nucleic-acid-tested versus non-nucleic-acid-tested communities in Arizona, U.S. scholars found that nucleic acid testing not only reduced the rate of transmission in the community but also substantially reduced mortality among COVID-19-infected individuals [35]. These studies considered the effects of isolation policy, economic policy, and health policy independently. Instead, we identify multiple combinations of factors that lead to the same phenomenon and explore governance pathways from a holistic perspective.

In summary, the existing literature contains conducted in-depth studies on the mechanism by which non-pharmaceutical intervention policies affect the spread of pandemics, revealing the pathways and factors influencing non-pharmaceutical intervention policies to effectively control the spread of pandemics. In fact, the effects of non-pharmaceutical intervention policies on controlling viral transmission are complex to evaluate, as they conceal cross-combination process, mutual influence, and joint action of multiple factors, which may not be isolated from each other but may have collaboration and interaction, so we estimate that the existing complex causal relationships between non-pharmaceutical intervention policies and pandemic prevention and control need to be examined from a holistic perspective. Early studies adopted a mathematical model of “one cause to one effect” or “multiple causes to one effect”, ignoring the interaction between antecedent variables, which can answer the mechanism of non-pharmaceutical intervention policies on pandemic control, but the explanatory power is insufficient. The QCA method reflects different combinations of dependent variables in the form of histories and uses case sample analysis as a qualitative comparison method of empirical evidence, which takes into account the interaction of conditional variables while getting rid of the limited thinking of correlation coefficients between variables. Looking for multiple combinations of factors leading to the same phenomenon from a broader perspective, the logical relationship of multiple concurrent causations [36] has the potential to provide solutions to the above-mentioned shortcomings. Improving the explanatory power of causality is more suitable for finding the best explanatory path for different outcome variables [37].

## 3. Methodology and Design

### 3.1. Research Methodology

In this paper, the fuzzy set qualitative comparative analysis (fsQCA) approach is used to investigate the problem of non-pharmaceutical intervention policies underlying multiple combinations mentioned earlier, as this method is known to be robust. FsQCA adheres to a holistic and systematic perspective by assuming the causes leading to the emergence of a phenomenon as different sets formed by a combination of variables and then translating complex cases into theoretical language through Boolean language and then exploring the causal mechanisms between the antecedent variables and the outcome variables [38]. Fuzzy set (fsQCA), clear set (csQCA), and multi-valued set (mvQCA) are the three mainstream analysis methods of QCA, among which fuzzy set qualitative comparative analysis stands alone as the best at dealing with case analysis problems that are not dichotomous, allowing for a partial affiliation between the antecedent and outcome variables from 0 to 1 [39]. Since both the antecedent and outcome variables selected in this paper have partial affiliation cases, it is more appropriate to use fsQCA.

### 3.2. Selection of Variables

#### 3.2.1. Selection of Condition Variables

In this paper, we used “novel coronavirus”, “COVID-19”, and “non-pharmaceutical intervention policy” as key words and “prevention and control measures, pandemic prevention and control, response measures, prevention and control strategies, protective measures, management strategies” as main themes. We collected related published articles on the Internet from 1 February 2020 to 31 January 2022 according to the degree of relevance, with the main themes of “prevention and control measures, pandemic prevention and control, response measures, prevention and control strategies, protective measures, management strategies” and the impact factor of the publications as the criterion. The top 100 articles were selected based on the impact factor of the publications; “COVID-19” and “non-pharmaceutical interventions” were used as the subject terms to screen the top 100 articles in the core set of the *Web of Science* papers. The top 100 articles were screened by the impact factor of publications, and the top 100 articles were screened by “COVID-19” and “non-pharmaceutical interventions”, and a total of 100 of the most highly cited and influential papers were screened.

We carried out a qualitative analysis of the collected and screened papers using NVivo platform. The 112 screened papers’ text was imported into NVivo 11 plus, and the information points appearing in the abstract, keywords, and conclusion of the papers were coded to nodes one by one by using word-by-word and sentence-by-sentence coding until the coded nodes had a high number of repetitions, and basically no new coded nodes appeared; then, the remaining uncoded text information was searched with the help of text query search function to ensure that no key was missed, coding as given in the node coding statistics table (Table 1) and node hierarchy diagram (Figure 1).

The above compilation of the literature shows that non-pharmaceutical intervention policies are developed by different sectors in response to a pandemic, which is in line with Mulford and Rogers’ definition of policy synergy: “the creation of new rules or the use of existing decision rules by two or more organizations that work together to respond to similar mission environments” as in the same vein. Policy synergies can be broadly classified as horizontal synergies that avoid conflicting policy objectives and ensure that individual policy supports each other, vertical synergies that ensure policy effects are consistent with policy-making intentions, and synergies in the time dimension that ensure policies continue to be effective. As it can be seen in Table 1, policies interact and influence each other.

The content of the paper was summarized through three stages of NVivo open coding, main axis coding, and selective coding, gradually conceptualizing and categorizing the textual information and finally summarizing the theoretical model of non-pharmaceutical intervention policies according to policy synergy theory. In Figure 2, the two-way arrows indicate that there is an interaction between the policies of different core categories, and the one-way arrows indicate that the policies of the three core categories work together to produce an effect, which in turn has an impact on the rate of viral transmission (Figure 2).

#### 3.2.2. Non-Pharmaceutical Policy Assignment Interventions

According to the constructed policy theory model, there are seven main categories of non-pharmaceutical intervention policies, such as work and school suspensions, assembly restrictions, mobility restrictions, home isolation, international population movement restrictions, income subsidies, and testing and screening. The identification and assignment of non-pharmaceutical intervention policies draw on the information from *Oxford COVID-19 Government Response Tracker* (Table 2).

#### 3.2.3. The Outcome Variables

In this investigation, the basic regeneration number is chosen as the outcome variable. The average number of people who can be infected by one patient is usually referred to as the basic regeneration number (R0) during the spread of a pandemic, and is one of the most important parameters used in academic research and epidemic control to determine whether an outbreak will be large-scale and to predict the trend of infectious diseases transmission [40]. Epidemiologists use the basic regeneration number (R0) assessment to predict the spread of the virus and to determine whether the spread of the pandemic can be effectively controlled under current measures so that non-pharmaceutical intervention policies can be adjusted in time.

#### 3.2.4. Computation Method

To assess the rate of viral transmission during the implementation of the policy, the SEIQR model was developed by adding isolation of susceptible compartments and isolation of confirmed compartments to the traditional susceptible-infective-removal (SIR) model, adding variables and parameters related to non-pharmaceutical interventions to better match the actual situation of viral transmission under the influence of non-pharmaceutical intervention policies. Based on the transmission mechanism of COVID-19 infectious disease and the actual situation of pandemic development, for the model to work and be accurate, we admit the following assumptions hold:

(1) Interacting individuals are divided into five groups or classes. Each group is represented in the model by its total population size at the considered unit of time. S group represents the susceptible individuals; E is the latent individuals or those who have been infected and are infectious but not showing symptoms; I formalizes the infected individuals, including asymptomatic and symptomatic infected ones; Q represents individuals who are quarantined after getting infected; R regroups all individuals who are immunocompetent after recovering from the disease;

(2) In our approach, we neglect migration, population movements, and birth, assuming they have insignificant effects in the period of crisis. Therefore, we set the total population size as constant: N = S + E + I + Q + R;

(3) Only “human-to-human” transmission is considered in this investigation, and we consider that isolated susceptible individuals are not involved in the spread of the pandemic;

(4) Once infected individuals are quarantined and isolated or confined in isolation, they are no longer able to transmit the disease;

(5) Recovered individuals do not develop immunity to the disease and may rejoin the susceptible group and could be re-infected.

The overall flow of the model is as follows: P proportion of the susceptible is involved in the spread of the pandemic through household isolation or self-isolation and wider pandemic prevention policies such as “city closures”. Contact is restricted between latently infected individuals, asymptomatic non-isolated individuals, asymptomatic infected individuals, and non-isolated susceptible individuals, resulting in susceptible individuals becoming, respectively, latent at β1 and β2 rates. Individuals in the incubation period become symptomatic at α rate. 1/α is the average incubation period for the disease. δ proportion of infected individuals are isolated, and this results in turning latent individuals into symptomatic ones, keeping them from moving to the infected susceptible individuals class. Infected individuals self-heal at γ1 rate, while 1-δ proportion of infected individuals who are not isolated can circulate and spread the virus. They self-heal naturally at γ2 speed with the possibility of becoming susceptible again.

Based on the above assumptions, the infectious disease transmission diagram was determined and illustrated (Figure 3).

The differential equations of the model are given in Equation (1) as follows:(1)$$\left\{\begin{array}{c} \frac{dS}{dt}=-\beta_{1}PSE-\beta_{2}PS\left(1-\delta \right)I+\left(1-\delta \right)\gamma_{1}I+\gamma_{2}Q\\ \frac{dE}{dt}=\beta_{1}PSE+\beta_{2}PS\left(1-\delta \right)I-\alpha E\\ \frac{dI}{dt}=\alpha E-\delta I-\left(1-\delta \right)\gamma_{1}I\\ \frac{dQ}{dt}=\delta I-\gamma_{2}Q\\ \frac{dR}{dt}=\left(1-\delta \right)\gamma_{1}I+\gamma_{2}Q \end{array}\right.$$

Based on (1), the basic regeneration number could be calculated using the next-generation matrix approach. The bin (E, I, Q) with infectious individuals was selected, and the analytical formula for the basic regeneration number was calculated as shown in Equation (2):(2)$$R_{0}=\rho \left(\dot{F}{\dot{V}}^{-1}\right)=\frac{\beta_{1}P}{\alpha }+\frac{\beta_{2}P}{\delta +\left(1-\delta \right)\gamma_{1}}$$

Using an optimal fitting solution, the infectious disease model is considered as a continuous system of nonlinear differential equations, the corresponding objective function is established, and the least squares method is applied to fit and estimate parameters value. Then, we substituted the obtained values into Equation (2) to compute the basic regeneration number of cases with values measuring the rate of viral transmission as the outcome variable of fuzzy set qualitative comparative analysis (fsQCA). The outcome variable (OUTCOME) with R_0_ > 1 was assigned a value of 0, meaning that the combination of non-pharmaceutical intervention policies was not effective in suppressing the pandemic; the outcome variable (OUTCOME) with R_0_ < 1 was assigned a value of 1, indicating that the combination of non-pharmaceutical intervention policies was able to suppress the pandemic. Seven categories of non-pharmaceutical interventions conditional variables, including work stoppage (C1), assembly restriction (C2), movement restriction (C3), home isolation (C4), international population movement restriction (C5), income subsidy (E1), and testing and screening (H1), were assigned values and integrated with two types of data, representing the underlying regeneration number of transmission rates, to form a sample of 102 cases.

## 4. Data Sources and Case Sample Selection

According to incomplete statistics, there are more than 6000 types of non-pharmaceutical intervention policies in the world, and it is necessary to select cases with representative and universal non-pharmaceutical intervention policies when selecting sample cases; otherwise, the results of the analysis would be meaningless from a practical point of view. In the selection of condition variables, we applied relevant theoretical criteria and identified a proportional relationship between variables and cases and selected 102 periods of stable, non-pharmaceutical intervention policies implementation from 233 countries and regions around the world.

The information used in the policy research is authoritative and available. The data are extracted from the University of Oxford open-source database of government policies in response to COVID-19 ((This database can be accessed at: https://github.com/owid/covid-19-data) accessed on 20 February 2022). The database contains collections of measures developed by national as well as provincial and state governments and provides a comprehensive and objective record of the evolution of non-drug policy interventions and the spread of the pandemic in each country.

## 5. Data Analysis and Empirical Results

### 5.1. Univariate Necessity Analysis

Univariate consistency and coverage calculation analyses were performed before conducting consistency analyses for different combinations of variables. Consistency analysis enables the examination of a single-variable necessity for the outcome, and the significance of univariate necessity analysis resides in assessing whether the conditional variable could allow us to reach a desirable output. Coverage represents the explanatory power of a single variable or combination of output variables, preventing a priori cognitive influences on the setting of explanatory variables. Generally, in the use of qualitative comparative analysis, X can be considered necessary for Y when the consistency is greater than 0.9.

FsQCA 3.0 was used to test whether each of the seven variables could be a necessary condition to inhibit the spread of the pandemic, and from the results (Table 3), it can be found that the consistency of all the variables did not reach 0.9, and the coverage values were high; therefore, these seven conditions did not meet a necessary condition for the results, proving that a single condition cannot effectively inhibit the spread of the pandemic but may be the result of a combination of variables acting together, which also validates the need for this investigation to study the effect of non-pharmaceutical intervention policies on the spread of the pandemic and its combination.

### 5.2. Analysis of Non-Pharmaceutical Intervention Policy Groupings

After necessity analysis for group analysis to explore multiple causal concurrent relationships of non-pharmaceutical intervention policies, this study used fsQCA 3.0 to conduct an empirical analysis of multiple concurrent influences on the calibrated outcome variables as well as the seven conditional variables. The conditional combination analysis according to fuzzy set qualitative comparative analysis (fsQCA) eventually yielded three results of parsimonious solutions covering all logical residuals, complex solutions that did not include logical residuals, and intermediate solutions that did not eliminate necessary conditions and did not deviate from case observations, as shown in Table 4. Most scholars believe that the intermediate solution incorporates the “logical residuals” that are consistent with theoretical and practical knowledge, and it combines the advantages of simple and complex solutions. For all the above reasons, we prioritized this study’s intermediate solution as the core component of our analysis with the ultimate goal of reaching more revealing and generalizable conclusions.

According to the fsQCA analysis rule, a condition that appears in both intermediate and simple solutions indicates that it has a significant influence on the result and is considered a “core condition” whose presence is indicated by ● and whose absence is indicated by ⊗. If a condition appears only in intermediate solutions but not in simple solutions, it has an auxiliary effect on the result. If a condition is present only in the intermediate solution but not in the simple solution, it is considered as a “marginal condition”, and its presence is denoted by ▲. If a condition is optional, it is considered as having a minor effect on the result and is denoted by “blank”. The results are shown in Table 5.

The four non-pharmaceutical interventions effective in suppressing the spread of the pandemic groupings are presented in Table 5, and the level of agreement between individual grouping and the overall grouping is above 0.8, which is higher than the minimum standard of 0.75. Among them, the consistency of the overall solution of the grouping of non-pharmaceutical intervention policies effective in suppressing the spread of the pandemic was 0.841, and the coverage was 0.547. It proves that the reasons for the combination of non-pharmaceutical intervention policies effective in controlling the spread of the pandemic are diverse and complex.

Four policy pathways of non-pharmaceutical interventions were found to be effective in controlling the spread of the pandemic. First, looking at the histories themselves, the policy combination for configuration L1a was “stop work and stop school * restrict assembly * ~ international population movement restrictions * income subsidy * testing and screening”, where restricting assembly, income subsidy, and testing and screening all played a central role, with stop work and stop school as the marginal conditions. This histogram has the highest consistency (0.83) and the highest raw coverage (0.481), with 20 countries, including the U.S., U.K., France, Turkey, Ireland, Spain, and Greece, fitting this scenario. The policy mix of the L1b configuration is “work stoppage * assembly restriction * movement restriction * home isolation * income subsidy, testing and screening”, where assembly restriction and income subsidy play the main role with testing and screening, while work stoppage, movement restriction, and home isolation are marginal conditions and play a supporting role in pandemic prevention and control. The consistency of the grouping was slightly lower than the grouping L1a (0.813), with a unique coverage of 0.022, with 17 countries, including Iceland, Singapore, Russia, and France, fitting this scenario.

The policy mix for configuration L2 is “work stoppage* ~ mobility restrictions * ~ home isolation * ~ international mobility restrictions * income subsidies * detection screening”, where work stoppage and detection screening play a central role, and income subsidies play a supporting role, with consistency (0.813) and unique coverage (0.026), with 20 countries such as Australia, Chile, Bangladesh, and Laos that could be used as case studies for this pathway. The policy mix for configuration L3 is “work stoppage * ~restriction of assembly * mobility restriction * ~home isolation * international mobility restriction * income subsidy * test screening”, with only one core element, international mobility restriction and work stoppage, mobility restriction, income subsidy, and test screening as marginal conditions that play supporting role. This grouping, while having the highest consistency (1), has the lowest unique coverage of the four paths (0.017), with only one country, China, available as a case study for this path. Second, in terms of individual conditions (cross-sectional), the three non-pharmaceutical intervention policies of stopping work and school, income subsidies, and testing and screening were present in all cohorts and played a universal role in effectively controlling the spread of the pandemic.

### 5.3. Discussion on the Four Non-Pharmaceutical Intervention Policy Pathways

To understand the specific effects of each non-pharmaceutical intervention policy pathway in different countries or regions, we discuss in detail the effects of each of the four pathways on controlling the spread of the pandemic in different countries or regions.

Configuration L1b indicates that the combination of the core conditions of restricting assembly, income subsidies, and testing and screening with the marginal condition of stopping work and school, movement restrictions, and home isolation effectively controls the spread of the pandemic by keeping the basic regeneration number below 1. Configuration L1a, on the other hand, indicates that the combination of marginal conditional work stoppage and school closure under the same core conditions is equally effective in controlling the spread of the pandemic. Comparing these two paths, it can be found that the group configuration L1b adopts two more non-pharmaceutical intervention policies of movement restriction and home quarantine than the group configuration L1a, both of which can effectively mimic the spread of the pandemic, from which it can be concluded that the group configuration L1b can first lift the movement restriction and home quarantine policies when relaxing the non-pharmaceutical intervention policies, open some factories and schools, and gradually restore the economic development and people’s daily life. L1a and L1b have the same core conditions of restricting assembly, income subsidies, and testing and screening, corroborating the significant role of restricting assembly, income subsidies, and testing and screening in controlling the spread of the pandemic. In contrast, the U.K.’s social isolation policy of requiring some schools and industries to suspend classes and work, banning rallies, closing public transport, advising home isolation, subsidizing income by more than 50% and reducing most debts, and screening suspected infected people with nucleic acid tests are non-pharmaceutical interventions that effectively have succeeded in controlling the spread of the pandemic. It is easy to see that the adoption of a high level of economic subsidies in conjunction with the shutdown of some industries has a huge social cost and a huge financial burden on the state. Although this grouping is effective in controlling the spread of the pandemic, it is less sustainable and can only delay the outbreak to a certain extent. If a vaccine or an effective drug cannot be developed in the meantime, it still cannot control the spread of the pandemic. Geographically, countries that have adopted this approach to prevent and control the spread of the pandemic are concentrated in Europe and in the Middle East. Therefore, to make effective non-pharmaceutical intervention policies, the possible duration of the pandemic and the social costs of policy implementation should be considered.

Configuration L2 has two core conditions of work stoppage and testing and screening and a marginal condition of income subsidy policy, suggesting that the spread of the pandemic can be controlled by adopting a work stoppage policy, strengthening testing and screening, and complementing it with an income subsidy policy. The main difference between this configuration and configuration L1 is that the income subsidy policy is only a marginal condition in configuration L2, suggesting that when the non-pharmaceutical intervention policy is relaxed in configuration L1a, priority can be given to weakening the income subsidy policy to reduce social costs.

With only one core condition, namely international population movement restrictions, and four marginal conditions, namely work stoppage, movement restrictions, income subsidies, and testing and screening, ConfigurationL3 has the highest consistency of the four pathways, but its original coverage is low, corresponding to only one country, China. This grouping is suitable when the pandemic has been largely controlled within the country, but the pandemic is still widespread in other countries, or it is adopted at the beginning of the pandemic to isolate the infected outside the country and reduce the number of imported cases, supplemented by the four measures of work stoppage, movement restriction, income subsidy, and detection screening, thus controlling the pandemic. When there are sporadic cases within the country, the spread of the pandemic can be better controlled by suspending work and classes in some areas and imposing traffic restrictions, strengthening testing and screening, and providing some income subsidies. This approach requires all active actors’ participation at all levels of the government, social groups, medical institutions, enterprises, and citizens and therefore requires a high level of political institutions and government credibility to prevent and control pandemics. Countries with low government credibility or countries with autonomous states (provinces) may not be able to achieve the desired results.

### 5.4. Robustness Tests

Robustness tests were conducted by adjusting the calibration quantile to draw relevant research results. This is done by adjusting “completely unaffiliated” (0.05) and “completely affiliated” (0.95) to “completely unaffiliated” (0.25) and “fully affiliated” (0.75), and the other steps remain unchanged. This outcome shows that core conditions and grouping paths do not substantially change, which proves the robustness of this research conclusions.

## 6. Conclusions

The following results were obtained from this study. (1) A total of four non-pharmaceutical intervention policy pathways were generated using fsQCA 3.0, which were as follows: L1, highly suppressive: L1a, “work stoppage * assembly restriction * ~ international population mobility restriction * income subsidy * detection screening”, and L1b, “work stoppage * restriction of the assembly * restriction of movement * isolation at home * income allowance, detection screening”; L2, moderately inhibited: “suspension of work and school * ~ restriction of movement * ~ isolation at home * ~ restriction of the international movement of the population * income allowance * detection screening”; and L3, externally inhibited: “ work stoppage * ~ assembly restrictions * movement restrictions * ~ home isolation * international population movement restrictions * income subsidies * testing screening”. (2) Individual non-pharmaceutical intervention policies are not independently effective in suppressing the spread of the pandemic. However, three types of non-pharmaceutical intervention policies, namely work stoppage, testing and screening, and financial subsidies, play a universal role in policy groupings effective in controlling the spread of the pandemic.

The existing literature has analyzed the effects of analyzing non-pharmaceutical intervention policies on the epidemic prevention and control effects from a single variable perspective, with less consideration of the interactions between variables. In this paper, we considered the multifactorial interactions of non-pharmaceutical intervention policies on epidemic prevention and control effects based on configuration perspective, analyzed the intrinsic relationships of each element, and then explored the conditional histology and pathways of non-pharmaceutical intervention policies for effective control of the epidemic transmission.

Results of our survey indicate that there is a need to reduce the cost of outbreak prevention and control and improve the sustainability of non-pharmaceutical intervention policies. A policy development group for non-pharmaceutical interventions needs to be established and adjusted according to the actual situation of the pandemic development. The intensity of policy disincentives varies among the four pathways, as evidenced by the different policy costs incurred when the pandemic can also be controlled. For this reason, on one hand, a non-pharmaceutical interventions formulation group was established to strengthen the capacity of scientific research and evaluation and gradually reduce the policy intensity and the impact on production and social life. On the other hand, the compliance of the people should be improved to ensure the sustainability of non-pharmaceutical intervention policies. From a global perspective, drug intervention measures should be systematically formulated, and economic subsidy policies should be implemented to reduce the economic pressure on enterprises and people, strengthen people’s compliance, and better implement social isolation policies, but practical economic subsidy policies should be formulated in accordance with the economic level and indebtedness of each country to avoid non-pharmaceutical intervention policies affecting normal operations of national economies.

It can be concluded from the study that opening up international population movements, complemented by policies such as testing and screening, information tracking, suspension from work and school, and movement restrictions, can effectively control the pandemic. Therefore, strengthening international cooperation, establishing an international platform for mutual recognition of pandemic information, and enhancing the tracking and monitoring of people moving across regions are effective measures to effectively counter the global spread of pandemic. On one hand, countries should strengthen cooperation to promote the establishment of information platforms and sharing of passage credentials, to identify potentially infected persons in a timely manner, and reduce interregional transmission of pandemics, based on which international trade can be launched; on the other hand, countries that have not implemented policies on international population movement restrictions should strengthen movement restrictions on incoming populations and improve testing and screening to provide guarantees for the continued opening of international population movement.

The construction and implementation of non-pharmaceutical intervention policies infrastructure should be strengthened, especially in areas with poor network infrastructure and weak testing and screening capacity, such as Yunnan, Qinghai, Tibet, and Ningxia (China), where the basic conditions for policy implementation are poor. After the promulgation and implementation of a series of non-pharmaceutical intervention policies, the construction of “basic conditions” should be strengthened to improve people’s compliance, to ensure that non-pharmaceutical intervention policies continue to effectively function, and to improve their effectiveness. For example, after the promulgation of the policy of work and school closure, the construction of online education and online office should be strengthened so that work will not be suspended, and school will not be stopped; after the promulgation of the policy of home isolation, the protection of people’s daily life should be strengthened to improve people’s compliance with the policy. Over-consuming countries need to adopt income-support policies to better support the implementation of interventions such as home isolation and work stoppage, while Asian countries with low personal indebtedness and high savings rates do not need to implement income-support policies to a great extent.

## 7. Limitations

With the development of globalization, an in-depth analysis of the relationship between non-pharmaceutical intervention policies and epidemic transmission is essential to reduce the waste of resources, stabilize economic growth, and protect life and health. This study answers the question of “the impact of non-pharmaceutical intervention policies in the spread of epidemics and the path of governance” as far as possible, but there are still some limitations.

First, we examined the effects of non-pharmaceutical intervention policies on the spread of the pandemic. However, we did not consider some relevant factors, such as different countries’ economic levels, their different medical levels, and their specific population age structures. Future studies could study more appropriate combinations of non-pharmacological intervention policies for different countries.

Next, our study on non-pharmaceutical intervention policies was conducted from a static perspective. In other words, because of the data limitations of the database, we examined the combinations of non-pharmaceutical intervention policies at a specific point in time. However, as the epidemic evolves, countries and regions may make policy adjustments. Studying the policy combination of non-pharmaceutical interventions at different points in time in various countries and regions can effectively help governments make policies to control the epidemic spread [41,42]. Future studies could the time-series changes in epidemic policies based on a dynamic perspective.

Finally, we studied the issue of policymaking from a macro perspective, while we did not study policy implementation in this article. If the citizens of a country do not follow the policy, the spread of the pandemic cannot be controlled as expected. Future studies could investigate how local governors can better implement the policy.

## Figures and Tables

**Figure 1 ijerph-20-00931-f001:**
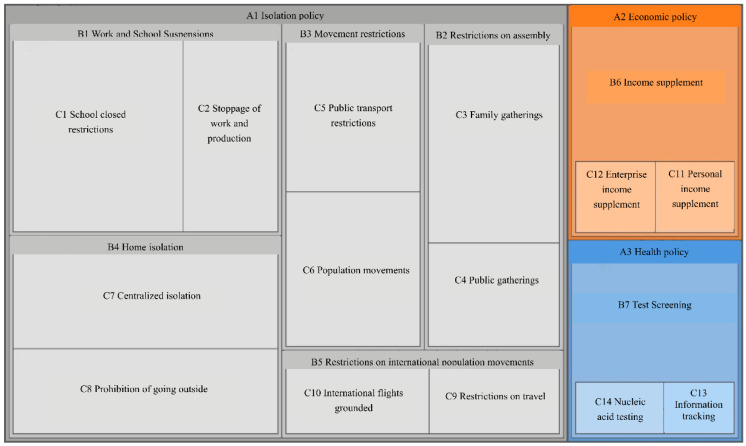
Node hierarchy diagram.

**Figure 2 ijerph-20-00931-f002:**
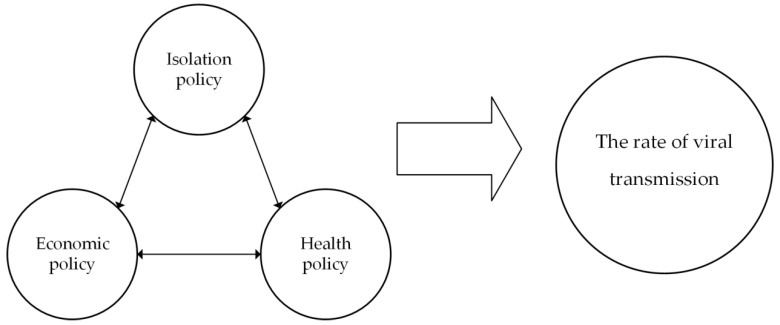
Diagram of non-pharmaceutical intervention policy for pandemic control.

**Figure 3 ijerph-20-00931-f003:**
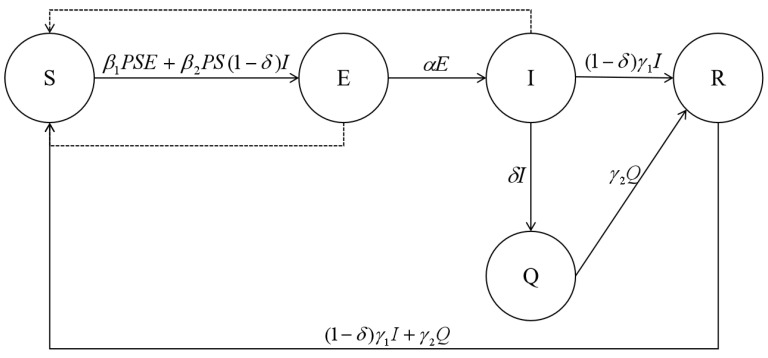
Flow chart of SEIQR dissemination.

**Table 1 ijerph-20-00931-t001:** Statistical table of node codes.

Core Category	Main Category	Initial Scope	Number of Reference Nodes	Number of Material Sources
Isolation policy (A1)	B1 Work and school Suspensions	C1 School closed	18	17
		C2 Stoppage of work and production	32	30
	B2 Restrictions on assembly	C3 Family gatherings	26	25
		C4 Public gatherings	13	13
	B3 Movement restrictions	C5 Public transport restrictions	20	20
		C6 Population movements	20	19
	B4 Home isolation	C7 Centralized isolation	25	24
		C8 Prohibition of going outside	22	22
	B5 Restrictions on international population movements	C9 Restrictions on travel	10	9
		C10 International flights grounded	11	11
Economic policy (A2)	B6 Income supplement	C11 Personal income supplement	6	6
		C12 Enterprise income supplement	6	6
Health policy (A3)	B7 Test screening	C13 Information tracking	4	4
		C14 Nucleic acid testing	5	3

**Table 2 ijerph-20-00931-t002:** Policy assignment for non-pharmaceutical interventions.

Name of Policy	Range of Values	Explanation of Values
Stop work and stop school C1	0	No work stoppage or school closure measures have been taken
	1	Suggest closing schools and workplaces or suggesting online classes and working from home
	2	Some school closures and industry shutdowns required
	3	A total shutdown of work and school is called for
Restrictions on assembly C2	0	No measures taken to restrict assembly
	1	Restriction of large gatherings or cancellation of social and public events gatherings
	2	Restrict larger gatherings (more than 10 people)
	3	Prohibition of assembly
Movement restrictions C3	0	No movement restrictions were applied
	1	Recommend closing public transportation or closing traffic routes; do not move across areas
	2	Require public transportation to be shut down (or banned for most citizens)
	3	Prohibition of inter-urban and intra-urban movement of persons
Home isolation C4	0	No home isolation measures were taken
	1	It is recommended not to go out
	2	Do not go out unless necessary. Do not go outside except for daily exercise, daily shopping, and necessary trips
	3	Prohibited from going outside except in special circumstances
Restrictions on international population movements C5	0	No measures have been taken to restrict the movement of people across borders
	1	Nucleic acid testing and isolation of entrants
	2	Ban on the entry of persons from certain regions and countries
	3	Ban on entry of persons from all countries and territories
Income supplement E1	0	No income supplement measures were taken
	1	Subsidies of less than 50 percent of income or only partial debt relief
	2	Subsidies exceed 50% of income and partial debt relief; subsidies are less than 50% of income and most debt relief
	3	Grants in excess of 50 percent of income and relief of most debts
Testing and screening H1	0	No testing screening measures were taken
	1	Only people who are symptomatic and meet specific criteria are tested. For example, medical personnel, exposed cases, returning from overseas
	2	Testing of all individuals exhibiting symptoms
	3	Non-discriminatory testing of all personnel

**Table 3 ijerph-20-00931-t003:** Results of univariate necessity analysis.

Conditional Variables	Effective Control of the Spread of the Pandemic	
	Consistency	Coverage
C1	0.713	0.777
~C1	0.287	0.705
C2	0.81	0.769
~C2	0.19	0.699
C3	0.555	0.781
~C3	0.445	0.725
C4	0.455	0.79
~C4	0.545	0.728
C5	0.117	0.9
~C5	0.884	0.739
E1	0.776	0.778
~E1	0.224	0.684
H1	0.690	0.816
~H1	0.310519	0.647

Note: “~” means not.

**Table 4 ijerph-20-00931-t004:** Results of fsQCA analysis.

Type of Result	Conditional Combination	Raw Coverage	Unique Coverage	Consistency
Complex solution	C1*C2*~C5*E1*H1	0.481	0.07	0.830
	C1*C2*C3*C4*E1*H1	0.299	0.022	0.813
	C1*~C3*~C4*~C5*E1*H1	0.298	0.026	0.813
	C1*~C2*C3*~C4*C5*E1*H1	0.03	0.017	1
	Solution coverage: 0.547			
	Solution consistency: 0.841			
Simple solution	C5	0.117	0.022	0.9
	C2*E1*H1	0.542	0.188	0.829
	C1*~C3*H1	0.354	0.008	0.813
	C1*~C2*~C3*E1	0.142	0	0.787
	Solution coverage: 0.607			
	Solution consistency: 0.834			
Middle solution	C1*C2*~C5*E1*H1	0.481	0.07	0.83
	C1*C2*C3*C4*E1*H1	0.299	0.022	0.813
	C1*~C3*~C4*~C5*E1*H1	0.298	0.026	0.813
	C1*~C2*C3*~C4*C5*E1*H1	0.03	0.017	1
	Solution coverage: 0.547			
	Solution consistency: 0.841			

Note: “~” means “not”; “*” means “and”; raw coverage means raw coverage; unique coverage means unique coverage.

**Table 5 ijerph-20-00931-t005:** Policy groupings for effective control of outbreak transmission.

Conditional Variables	Policy Configurations for Effective Control of the Spread of the Pandemic			
	L1a	L1b	L2	L3
Suspension of work and school (C1)	▲	▲	●	▲
Restrictions on assembly (C2)	●	●		⊗
Movement restrictions (C3)		▲	⊗	▲
Home isolation (C4)		▲	⊗	⊗
Restrictions on international population movements (C5)	⊗		⊗	●
Income supplement (E1)	●	●	▲	▲
Test screening (H1)	●	●	●	▲
coherence	0.83	0.813	0.813	1
Raw coverage	0.481	0.299	0.3	0.03
Unique coverage	0.07	0.022	0.026	0.017
Consistency of solutions	0.841			
Coverage of solutions	0.547			

Note: ● means core condition exists; ▲ means edge condition; ⊗ means inexistent.

## Data Availability

Data presented in this study are available upon request to the corresponding authors.

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
