# Peer review of "Multiple Concurrent Causal Relationships and Multiple Governance Pathways for Non-Pharmaceutical Intervention Policies in Pandemics: A Fuzzy Set Qualitative Comparative Analysis Based on 102 Countries and Regions"

_ijerph, 2023, doi:10.3390/ijerph20020931_

Round 1

Reviewer 1 Report

Dear all,

General aspects

The article utilizes a good strategy and outlines crucial aspects about how non-pharmaceutical intervention policies prevent and control the spread of pandemics from the configuration perspective.

The article's organization follows a logical and consistent pattern. The results produced in accordance with the method adopted lead to the conclusions.

The majority of the references are recent publications.

 Specific aspects

1.      1. Please include some information’s about of limitations of study.

 2.      Also is necessary to add a short subchapter regarding discussion. Here may be is important to discuss about four non-pharmacological intervention policy pathways: two highly suppressive, moderately inhibited, and externally inhibited

 Best regards,

Reviewer 2 Report

1.   What is the main question addressed by the research?

This study analyzes how non-pharmaceutical intervention policies prevent and control the spread of pandemics from the configuration perspective. Using fuzzy set qualitative comparative analysis method to analyze data from a sample of 102 countries and regions around the world, seven representative conditional variables are selected to explore the conditional histories and pathways of non-pharmaceutical intervention policies with effective control of pandemics spread.

2.   Do you consider the topic original or relevant in the field? Does it address a specific gap in the field?

The spread of the COVID-19 virus is a complex problem felt around the world. Global tracking and control of the virus requires not only pharmaceutical intervention but also the widespread use of non-pharmaceutical policies.

Finding effective policies (or even, as the Authors pointed out, "a combination of policies") for non-pharmaceutical interventions to deal with the spread of pandemics is an important challenge for social development and therefore makes the topic of this study extremely relevant.

3.   What does it add to the subject area compared with other published material?

The Authors argue that the complex causal relationship between non-pharmaceutical intervention policies and pandemic prevention and control need to be examined from a holistic perspective. The Authors are convinced that non-pharmaceutical intervention policies that affect on controlling viral transmission conceals cross-combination process, mutual influence and joint action of multiple factors, which may not be isolated from each other but have some kind of collaboration and interaction.

As a result of their research, the Authors presented three types of non-pharmacological intervention policies that play universal role in policy groupings effective in controlling the spread of the pandemic.

4.   What specific improvements should the authors consider regarding the methodology? What further controls should be considered?

The manuscript is clearly structured. The methodology used in the study is not objectionable to me.

5.   Are the conclusions consistent with the evidence and arguments presented and do they address the main question posed?

The aims stated by the Authors have been achieved. Conclusions are drawn correctly and correspond to the content of the conducted scientific analyzes.

6.   Are the references appropriate?

The literature cited in the work corresponds to the current research status in the field of the discussed issues.

7.   Please include any assitional comments on the tables and figures.

No critical remarks regarding tables and figures.

8.   Additional comments or continuations of above sections.

In line 241 source reference 40 is given as superscript.

Reviewer 3 Report

This paper investigates the effectiveness of non-pharmaceutical intervention policies on the spread of Covid-19 throughout a SEIQR model and a fsQCA method. The study is well designed, with appropriated Computation method. The conclusions are convincing and seem trustable. Thus, the whole manuscript is perfectly correct.
